# Influence of a Calcium Phosphate Coating (BONIT^®^) on the Proliferation and Differentiation Potential of Human Mesenchymal Stroma Cells in the Early Phase of Bone Healing

**DOI:** 10.3390/jfb13040176

**Published:** 2022-10-06

**Authors:** Andy Eggert, Bettina Alexandra Buhren, Holger Schrumpf, Marcel Haversath, Martin Ruppert, Marcus Jäger, Rüdiger Krauspe, Christoph Zilkens

**Affiliations:** 1Clinic for Orthopaedic Surgery, Heinrich-Heine-University, Moorenstraße 5, 40225 Düsseldorf, Germany; 2Department of Arthroscopy and Endoprosthetics, Krankenhaus Nettetal GmbH, Sassenfelder Kirchweg 1, 41334 Nettetal, Germany; 3Clinic for Orthopaedic Surgery, St. Marien-Hospital Mülheim an der Ruhr, Kaiserstraße 50, 45468 Mülheim an der Ruhr, Germany; 4Orthopaedics, Trauma and Reconstructive Surgery, University of Duisburg-Essen, Forsthausweg 2, 47057 Duisburg, Germany

**Keywords:** calcium phosphate, osseointegration, stem cells, bone metabolism, gene expression, orthopedic implants, surface

## Abstract

When implanting osteosynthetic materials or orthopedic implants, the surface condition plays a decisive role for mid- to long-term osseointegration. BONIT^®^, an electrochemically produced calcium phosphate (CaP) coating, has been used in the surface refinement of implants since 1995. More than 3.5 million coated implants have been successfully placed so far. BONIT^®^ has thus been able to demonstrate clinical success. However, due to its surface properties and solubility, and the resulting difficulty in culturing cells, there are no in vitro studies investigating its influence at the molecular level, particularly on bone metabolism. In a first step, the cells from a total of ten donors were seeded separately on four different surfaces: 1. a pure corundum-blasted titanium surface (CELLTex^®^, CT), 2. CT with additional BONIT^®^ coating (CT + B), 3. a hydroxyapatite-blasted titanium surface (DUOTex^®^, DT), 4. DT with additional BONIT^®^ coating (DT + B). In a second step, the cells were grown for 48 h. The proliferation behavior and differentiation potential of hMSCs were investigated at three consecutive time points (12 h, 24 h and 48 h) by quantifying the mRNA expression of ten important differentiation markers using quantitative real-time polymerase chain reaction (qRT-PCR). We were able to show that BONIT^®^ has an influence on the early proliferation and differentiation behavior of hMSCs in patients of all age groups. The additional BONIT^®^ coating on CELLTex^®^ or DUOTex^®^ led to a defined mRNA expression pattern for the investigated factors: a tendency towards a higher expression rate with coating present could be found for bone morphogenetic protein 2 (BMP2), osteopontin (OPN), osteocalcin (OC), receptor activator of NF-κB ligand (RANKL) and osteoprotegerin (OPG). A similar or lower expression rate was detected for runt-related transcription factor 2 (RUNX2), alpha-1 type I collagen (COL1A1), alkaline phosphatase (AP), osteonectin (ON) and insulin-like growth factor I (IGF1). We have developed a new method that allows the cultivation of human mesenchymal stromal cells (hMSCs) on the soluble coating BONIT^®^ for gene expression analysis. BONIT^®^ has a significant influence on the proliferation and differentiation behavior of human mesenchymal stroma cells. This study describes a defined gene expression pattern of bone metabolism that may help to understand the influence of this CaP coating on the early phase of implant osseointegration.

## 1. Introduction

Artificial joint replacement is a globally successful procedure for the treatment of degenerative joint diseases. The World Health Organization estimates that the number of bone and joint diseases has doubled in the last 20 years and degenerative diseases are the global leader in this field [1,2]. Diseases of the musculoskeletal system are not only debilitating with regard to physical complaints, such as pain or functional restrictions; they also prevent participation in social life and are an increasing financial burden for the health care system [3,4,5,6]. The clinically successful procedure of artificial joint replacement restores the mobility of the joints and relieves the pain of the affected patients.

However, the survival rate of joint replacement is limited and the main causes for implant failure are still septic and aseptic loosening. The failure rate increases with time after primary surgery. Therefore, younger patients are at higher risk of undergoing revision of the implant based on their life expectancy alone [1,2]. The replacement of an implant is associated with additional pain and risks and may result in a reduction in quality of life. If the revision frequency can be reduced by 1%, costs of more than 430 million euros could be saved [7].

To overcome the limitation in durability, much effort has been directed toward material improvement in recent decades through the avoidance of wear or enhancement of the capability of osseointegration. Of course, the implantation technique forms the basis for good healing of the implant in the bone [7].

In cementless fixation of implants, a strong bond (press fit) between the bone and the implant is very important to promote stability through bone growth [8,9]. Different materials are used for manufacturing of prosthesis in order to fulfil this requirement [10,11]. The material of the implants must be fracture-resistant and biocompatible over a long period of time [12]. While the special shapes and materials of prostheses have proven their worth and have been well investigated, more and more work is being undertaken on surface structures that could enable faster and better bone ingrowth [13,14]. Hydroxyapatite, a calcium phosphate ceramic, has been one of the most common and best-analyzed surface coatings since 1970 [12,15]. Calcium phosphates have ideal biocompatibility due to their composition [16]. In addition, other calcium phosphate ceramics, oxide ceramics and glass ceramics are also used in the surface refinement of titanium or stainless steel-based materials [12,17,18].

A widely used process for the production of hydroxyapatite coatings is the plasma-spraying process [12]. Even if better osseointegration takes place under plasma-sprayed hydroxyapatite coatings [9,19], there are several disadvantages resulting from the extended manufacturing process. One example is the decomposition of hydroxyapatite (HA) by thermal treatment due to the catalytic effect of the base material. Robert B. Heimann assumes that this decomposition takes place up to the surface of the coating, resulting in poorer resistance [20].

BONIT^®^ is a calcium phosphate (CaP) coating developed by the company DOT GmbH that has been used since 1995 in the surface refinement of implants [21]. The BONIT^®^ coating involves the electrochemical deposition of a thin, bioactive calcium phosphate layer [22]. To date, more than 3.5 million implants have been implanted in patients of different ages with BONIT^®^ [21]. As part of the approval process for medical devices, numerous physical tests and animal experiments have been carried out with BONIT^®^ coatings. These tests have shown advantages over the conventional hydroxyapatite coating with regard to the capacity for osseointegration [22,23,24,25,26,27,28,29,30]. Furthermore, BONIT^®^ has proven clinical efficiency [31,32]. Currently, there are no in vitro studies that have successfully investigated the influence of BONIT^®^ at the molecular level, particularly at the level of gene expression of bone metabolism in the early phase of bone healing. Up to now, the cultivation of cells has only been possible to a limited extent. An in vitro study described how osteoblast cells applied to BONIT^®^ are covered by a precipitate within 48 h [33]. Another in vitro study showed that proliferation of BONIT^®^ does not occur during the first weeks. Hentschel investigated the proliferative impact of unrestricted somatic stem cells (USSCs) in relation to the surface roughness and chemical properties of BONIT^®^ on cell proliferation [34]. There is still a gap in research on the in vitro behavior of cells seeded on this coating. Based on the present results, we developed a new technique for seeding and growing hMSCs in vitro on test specimens and focused on the relative mRNA expression of their key differentiation markers in the early phase of bone healing.

## 2. Material and Methods

The study protocol was approved by the Ethics Committee of the Medical Faculty of Heinrich Heine University Düsseldorf in accordance with the Helsinki Declaration (Project identification code: 5401). Written informed consent to participate in this study was obtained from all patients prior to surgery. During elective surgery, bone marrow was removed from the femoral (five female and five male cell donors; average age was 45.89 years (SD ± 26.51 years)).

Four different surface refinements were tested (all from DOT GmbH, 18059 Rostock, Germany): 1. CELLTex^®^ (CT), a pure titanium surface generated through a corundum particle-blasting process and acid etching (roughness 3.0 ± 1.5 µm); 2. CELLTex^®^ with additional BONIT^®^ CaP-layer (CT + B); 3. DUOTex^®^ (DT), a hydroxyapatite (HA)-blasted and double acid-etched subtractive surface (roughness 1.1 ± 0.5 µm); and 4. DUOTex^®^ with additional BONIT^®^ CaP-layer (DT + B). BONIT has a layer thickness of 20 ± 10 µm. The rounded sample plates had dimensions of 11.5 mm in diameter and 2 mm in thickness (Figure 1). The total surface of each sample was 6.23 cm^2^.

Cell isolation: We did not use a defined cell line. The source of our mesenchymal stem cells (hMSC) was a mixture of spongiosa and bone marrow. This individual mixture from each patient was used for the following cell separation and seeding procedure with all of the four different sample plates.

The mononuclear bone marrow cells were separated by density gradient centrifugation (Biocoll 1077 g/mL, Biochrom GmbH, Berlin, Germany) and converted to DMEM (Sigma-Aldrich, St. Louis, MO, USA) with 20% fetal bovine serum (FBS, FBS Superior, Biochrom GmbH) and 1% penicillin/streptomycin/L-glutamine (PSG, Sigma-Aldrich) in a humid atmosphere at 37 °C and 5% CO_2_ in tissue culture bottles (CELLSTAR^®^, Greiner Bio-One GmbH, Frickenhausen, Germany). Non-adherent cells were removed after five days. The procedure fulfilled the first property of the minimal criteria to define hMSCs according to Dominici et al. [35]. Further characterization of the cells did not take place in this study. However, we refer to the successful use of this technique by our laboratory [36]. A medium change was performed alternately every three to four days. The cells were passed weekly using trypsin-ethylenediaminetetraacetic acid (trypsin-EDTA, Sigma-Aldrich, St. Louis, MO, USA), re-sown and incubated at a density of 5000 cells per cm^2^. All hMSC donor cells were counted in passage two and placed in a cryotube (CRYO.STM, Greiner Bio-One GmbH, Frickenhausen, Germany) into freezing medium (fetal bovine serum) with 10% dimethyl sulfoxide (DMSO, Sigma-Aldrich, St Louis, MO, USA) and preserved in liquid nitrogen (MVE Cryosystem 4000, MVE Inc., 3505 Country Road, 42 W-Burnsville, MN 55306, USA).

Experimental setup: In this study, each surface group (1–4) was tested twice with the cells of every single patient. The investigated kinetic points were determined at 12, 24 and 48 h. A total of 75,000 donor cells were applied per test plate. The passage two cells were removed from the cryopreservation, defrosted and transferred into 50 mL fresh DMEM medium with 20% FBS and 1% penicillin/streptomycin/L-glutamine. The cells were centrifuged at 1800 rpm at room temperature for five minutes. The supernatant was decanted and discarded. The cell pellet was resuspended with 1 mL medium. The cell count was carried out using a Neubauer counting chamber. The sample plates were transferred sterile into the medium wells of a 24-well plate. The surfaces to be tested were aligned skyward. The outer corrugated chambers were filled with 1 mL medium to achieve sufficient humidity in the corrugated chambers and to compensate for temperature fluctuations. A total of 80 µL medium containing 75,000 cells was applied as a standing drop to the sample plates. The corrugated sheets were incubated for 90 min at 37 °C and 5% CO_2_. We equipped eight wells of a 12-well plate with an incubation aid made of stainless steel wire (Figure 1) and flooded the wells with 4300 µL medium. The specimens were removed from the chambers of the 24-well plate with sterile forceps. The standing drop was poured off. The plates were turned while floating by 180° and placed on the incubation aid.

The surface to be tested with the adherent cells pointed towards the bottom of the corrugated plate and was laid completely in the medium. This was followed by incubation at 37 °C and 5% CO_2_. The cells were incubated for a further 10.5 h (±0.5 h) for the 12 h kinetic point, 22.5 h (±0.5 h) for the 24 h kinetic point and 46.5 h (±0.5 h) for the 48 h kinetic point. The sample plates were removed from the incubation aid with sterile forceps, turned 180° and washed once in phosphate-buffered salt solution (PBS, Sigma-Aldrich, St. Louis, MO, USA). After washing, the cells were taken up in 360 µL RLT buffer (Qiagen, Hilden, Germany) with 10 μL/mL 2-mercaptoethanol (Sigma-Aldrich, St. Louis, MO, USA) and frozen at −80 °C. The cells were then stored in a freezer. Figure 2 shows the schematic sequence of the test series.

Quantitative real-time PCR: The mRNA of the ten donors was isolated according to a standardized procedure. The RNeasy Mini-Kit 250 and the RNase-free DNase set (both Qiagen, Hilden, Germany) were used for isolation. This was followed by a standardized transcription of the mRNA into cDNA. For this research, the QuantiTect^®^ Reverse Transcription Kit from Qiagen was used for the synthesis of cDNA according to the instruction for use.

Furthermore, a StepOne real-time PCR system (StepOne^TM^, software version 2.2.2) with SYBRTM Green Master Mix (both from Applied Biosystems, Life Technologies, Carlsbad, CA, USA) was used for qRT-PCR. Gene expression analysis was performed after the addition of primers from Qiagen or Biolegio (Nijmegen, the Netherlands). Glycerinaldehyde 3-phosphate dehydrogenase (GAPDH) served as a reference control. The relative gene expression levels of bone morphogenetic protein-2 (BMP2), runt-related transcription factor 2 (RUNX2), alpha-1 type I collagen (COL1A1), alkaline phosphatase (AP), osteopontin (OPN), osteonectin/secreted protein acidic and rich in cysteine (ON), osteocalcin/bone γ-carboxylglutamic acid-containing protein (OC), insulin-like growth factor 1 (IGF1), receptor activator of NF-κB ligand (RANKL) and osteoprotegerin (OPG) were calculated using the 2^−^^ΔΔCt^ method [37].

Statistical evaluation: The results were recorded and the relative units calculated using Excel Office 365 (Microsoft Corporation, Redmond, WA, USA). The statistical evaluation was carried out with GraphPad Prism 7 (GraphPad Software Inc., San Diego, CA, USA).

CELLTex^®^ (CT) was compared with CELLTex^®^ + BONIT^®^ (CT + B) and DUOTex^®^ (DT) with DUOTex^®^ + BONIT^®^ (CT + B), respectively. A Mann–Whitney U test was performed to compare the relative units (RU). A paired parametric *t*-test was performed to analyze the temporal course of gene expression on a coating. The following levels of significance were assigned to the figures: * *p* ≤ 0.05 (significance), ** *p* ≤ 0.01 (high significance) and *** *p* ≤ 0.001 (very high significance).

## 3. Results

### 3.1. Genes with a Tendency for Decreased or Similar Expression with BONIT^®^

RUNX2: This key transcription factor associated with osteoblast differentiation could be detected at all kinetic time points and on all four surface structures. After 12 h of incubation, the additional BONIT^®^ CaP-Layer led to a significant reduction of RUNX2 expression compared to the non-refined CT or DT surfaces (*p* ≤ 0.05; Figure 3a). This initially lower expression nearly and considerably approached the values of the CaP-untreated surfaces up to 48 h, with no more significant differences between 24 h and 48 h. The expression on both BONIT^®^ surfaces increased significantly over time, while on the CaP-untreated surfaces (CT and DT) there was a relatively significant decrease from 12 h to 24 h (*p* ≤ 0.05 for CT; *p* ≤ 0.01 for DT) and then a slight increase again up to 48 h.

COL1A1: Expression of the proteine-coding gene COL1A1 was found on all specimens. After 12 h of incubation, no significant differences in expression could be found when comparing the four different surfaces. However, after 24 h and 48 h, BONIT^®^ resulted in a partially significant difference in terms of a lower expression rate compared to the untreated surfaces (CT vs. CT + B at 24 h, *p* ≤ 0.05, and at 48 h, *p* ≤ 0.01; DT vs. DT + B at 48 h, *p* ≤ 0.05). The expression patterns over time on each surface showed relative increases of COL1A1, which were more evident for the non-layered CT and DT surfaces (Figure 3b).

AP: Alkaline phosphatase is a reliable marker for bone metabolism as it dephosphorylates compounds. The expression of the AP gene did not differ significantly after 12 h when comparing the four surfaces. However, between 12 and 24 h incubation time, a (highly) significant difference was found between the surfaces: with BONIT^®^, the expression of AP was siginficantly lower than with the non-layered samples, and this applied to CT vs. CT + B, as well as DT vs. DT + B, respectively (after 48 h, *p* ≤ 0.01). Comparing the expression of the gene over time, almost all surfaces showed a peak after 24 h, followed by a drop after 48 h (Figure 3c). The expression tended to be significantly higher with the untreated surfaces, as already described above.

ON: Osteonectin, also known as secreted protein acidic and rich in cysteine, initiates mineralization and promotes mineral crystal formation during bone formation. The gene expression analysis of ON showed exclusively lower values for both surfaces with BONIT^®^ at all kinetic time points(Figure 3d). In the DT group, the expression of the cells on with added BONIT^®^ coating was significantly lower at 12 h compared to DT alone (*p* = 0.0355). After 48 h of incubation time, the difference for the lower AP expression with BONIT^®^ was highly significant in both comparisons: CT vs. CT + B (*p* = 0.0015) and DT vs. DT + B (*p* = 0.0011). Over time, the non-coated surfaces showed a steady increase in the expression of ON, while the samples with the BONIT^®^ coating showed a peak at 24 h. With the control surfaces CT and DT, the expression of ON was significantly upregulated from 24 h to 48 h (CT: *p* = 0.0041; DT: *p* = 0.0098).

IGF1: The gene expression of the anabolic protein IGF1 showed similar values for all four surfaces at all time points (Figure 3e). On the other hand, the following picture emerged when looking at the time course with each individual surface: at 12 h, a very low expression rate was detectable, which increased sharply up to 24 h and then dropped again moderately. These differences were, for the most part, statistically significant: 1. the increase from 12 h to 24 h (CT, *p* = 0.021; CT + B, *p* = 0.0188; DT, *p* = 0.0129; DT + B, *p* = 0.0069), and 2. the decrease from 24 h to 48 h (CT + B, *p* = 0.0451; DT + B, *p* = 0.0142). After 48 h of incubation, the IGF1 expression was still significantly higher than at 12 h (CT, *p* = 0.0003; CT + B, *p* = 0.0087; DT, *p* = 0.0007; DT + B, *p* = 0.0033).

### 3.2. Genes with a Tendency for Increased Expression with BONIT^®^

BMP2: The gene expression analysis of the bone formation inducing BMP2 showed major differences between the surfaces (Figure 4a). Cells on BONIT^®^ showed higher expression of the BMP2 gene at all time points. Significant difference were observed from the 24 h time point (at 24 h: CT vs. CT + B, *p* = 0.04; DT vs. DT + B, *p* = 0.04; at 48 h: CT vs. CT + B, *p* = 0.004; DT vs. DT + B, *p* = 0.0019).

The highest values for BMP2 expression appeared at 12 h, which then dropped again continuously and significantly with all four surfaces over time. An overall higher level could be detected for the BONIT^®^ surfaces. The decrease from hour 12 to hour 48 was highly significant in all cases (12 h vs. 48 h: CT, *p* = 0.0055; CT + B, *p* = 0.005; DT, *p* = 0.0081; DT + B, *p* = 0.0016).

OPN: Expression of the hydroxyapatite-binding matrix protein osteopontin was detected on all surfaces. Within the first 24 h, there were no significant differences in the expression of OPN between the surfaces (Figure 3b), whereas, at 48 h the kinetic point, a significant increase was found for the BONIT^®^ surfaces compared to the native CELLTex and DUOTex surfaces (at 48 h: CT vs. CT + B, *p* = 0.0115; DT vs. DT + B, *p* = 0.0052). A similar time course over 48 h was observed for the expression of OPN for all surfaces. From hour 12 to hour 24, there was a similar expression of OPN at a low level, and CT even showed a significant decrease (*p* = 0.0336). From hour 24 to hour 48, the expression for all surfaces, including the BONIT^®^-coated surfaces, was significantly to highly significantly upregulated (24 h vs. 48 h: CT, *p* = 0.0218; CT + B, *p* = 0.0016; DT + B, *p* = 0.0177). This effect was less pronounced for the native DT surface.

OC: The gene expression of the matrix protein OC was found with every surface (Figure 4c). The expression pattern was similar to that of osteopontin. Within the first 24 h, only minor differences were observed, but, after 48 h, the average expression values for the BONIT^®^ surfaces were more than twice as high. This difference was found to be very highly significant (at 48 h: CT vs. CT + B, *p* = 0.0003; DT vs. DT + B, *p* = 0.0005). The slight tendency toward higher OC expression with BONIT^®^ at hours 12 and 24 was not significant. A similar time course over 48 h was observed for the expression of OC for all surfaces. From hour 12 to hour 24, no increase in expression was observed. From hour 24 to hour 48, the expression reached the highest values with all four surfaces. The BONIT^®^ surfaces showed a particularly high increase (24 h vs. 48 h: CT + B, *p* = 0.0020; DT + B, *p* = 0.0031).

RANKL: The gene expression analysis of the osteoclast-regulating RANKL revealed higher values with BONIT^®^ at all kinetic time points (Figure 4d). This difference was very pronounced after 48 h. The expression of RANKL with the CT + B surface was increased at 48 h compared to CT with a significance value of *p* = 0.0039, and this was similar to DT + B vs. DT with *p* = 0.0185, respectively. The time course showed a similar pattern with all surfaces, and generally (except with CT + B) a peak at 24 h was visible. Significant differences could only be detected with the BONIT^®^ coatings. CT + B exhibited a significant upregulation of RANKL from hour 12 to hour 48 (*p* = 0.0109). For DT + B, a significant upregulation from 12 to 24 h (*p* = 0.0164) was observed.

OPG: The expression of the osteoclastogenesis inhibitory protein osteoprotegerin was detected for all surfaces (Figure 4e). More OPG was expressed with the BONIT^®^-refined surfaces: at hour 12, the level of significance for CT vs. CT + B was *p* = 0.0039, and for DT vs. DT + B, it was *p* = 0.0288, respectively. In relative comparison, the values between the high BONIT^®^ and the low natives were farthest apart at the beginning, while they converged towards 48 h. However, even at hour 24, the difference in the higher OPG expression with BONIT^®^ had a relevant significance level (CT vs. CT + B, *p* = 0.0185; DT vs. DT + B, *p* = 0.0083; the same was found at hour 48). A similar time course over 48 h was visible for all surfaces, with high expression at the beginning, a minimum at 24 h and, again, high expression at the end of the observation period. From hour 12 to hour 24, the expression rate decreased for all surfaces. Three conditions showed significant to very highly significant decreases (12 h vs. 24 h: CT, *p* = 0.0171; CT + B, *p* = 0.001; DT + B, *p* = 0.0053). From hour 24 to hour 48, OPG expression increased significantly for DT (*p* = 0.0355).

## 4. Discussion

The search for the ideal temporal expression of proteins for perfect, long-term, stable osseointegration of implants remains exciting. It is known that not only the implant material itself but also the surface structures at the micro-, nano- and sub-nano-scales have a strong influence on adherent cells in the bone, especially in the early phase [38]. Up to 50% by volume and 70% by weight of the composition of bone is comprised of a modified form of a calcium phosphate compound called hydroxyapatite [39]. Calcium phosphate compounds have been used successfully for many years as a bone substitute material in different forms, from granulat to fast-curing cements, all bioresorbable. It is therefore more than logical that, in the development and refinement of implant surfaces, attempts have been made to use substances similar to bone substance to improve the healing process after surgical implantation [40,41]. BONIT^®^ is a successful attempt that has made it to market and beyond and has now been clinically used over 3.5 million times since 1995. The BONIT^®^ coating is an electrochemical deposition of a thin, bioactive calcium phosphate layer and is a composite of bruschite and hydroxyapatite, which is characterized by very good biocompatibility [23]. The molar ratio of calcium to phosphate is 1.1 ± 0.1 [24]. BONIT^®^ already provides a high concentration of calcium and phosphate ions in the solution phase at the outset and it occurs as an intermediate in natural bone itself [42]. Proportionally, hydroxyapatite is slowly absorbed over a period of 6 weeks and continuously releases ions [22].

The coating of dental implants, but also of orthopedic implants (e.g., shaft endoprostheses with the Hofheim model, intervertebral endoprostheses since 1998, Pressfit cups since 2004), is increasing. Negative feedback from post-market surveillance is not known [33]. Clinical observations, including femoral shaft arthroplasties, intervertebral arthroplasties, meta-short stem arthroplasties and press-fit cups coated with BONIT^®^, show good and faster healing behavior in follow-up studies [43,44,45,46]. For example, Mockwitz et al. described an overall loosening rate of 1.4% and a HARRIS hip score of 88.53% for cementless femoral stem arthroplasty [43].

Szmukler-Moncler et al. showed in animal experiments that BONIT^®^ has an osteoconductive effect, especially in the cancellous part of the bone, and suggested that this is also due to the rough, porous surface [22]. In his work, Hentschel demonstrates a surface roughness (Ra) for BONIT^®^ of 1.23 µm [34]. In the established literature, it is known that surface roughness has a significant influence on cell behavior (adhesion, proliferation, differentiation). However, a direct comparison between the different studies is difficult because not only the surface roughness of the base material but also the chemical behavior, phase inventory, porosity, crystallinity, layer thicknesses and underlying cell line strongly influence the results. Furthermore, it has to be considered that an in vitro experiment represents a self-contained system.

In this in vitro study, we compared four different constellations with different properties. Here, CT exhibited a roughness (Ra) of 3.0 ± 1.5 µm and DT a roughness (Ra) of 1.1 ± 0.5 µm. In contrast to CT and DT, BONIT^®^ itself dissolves in aqueous medium. It should be noted that, with soluble coatings, loss of cell material can occur not only during medium change but also during cell isolation. This may also have influenced the results in this study.

Hentschel showed that there were significantly fewer cells (USSCs) on the very rough surface of BONIT^®^ than on the smoother comparison surfaces. However, the cell number increased with a decrease in roughness [34]. Toxic properties of the BONIT^®^ coating could be excluded by the Bioservice Scientific Laboratories GmbH, 82152 Planegg, Germany (Study no. 101981.2010).

Another in vitro study, also published by DOT GmbH, dealt with the precipitation behavior of BONIT^®^. The osteoblast cell line MG-63 was cultured for 48 h on a BONIT^®^ sample. It was found that raised cells were almost completely covered with a fine crystalline calcium phosphate precipitate after only 48 h [33]. This observation led us to attempt upside-down cultivation to make the cells accessible to gene expression. It is the material and the surface of the implant that determine how the local cells in the bone react to the implant. Mesenchymal stromal cells play an important role in the regeneration and healing of tissues [47]. They are able to develop into different tissues [48]. Mesenchymal stromal cells (MSCs) can differentiate into osteoblasts under certain conditions [49]. The local environment of these cells has an influence on subsequent differentiation. Stromal cells from the periosteum increasingly differentiate into chondrocytes and osteoblasts, and cells from the medullary cavity increasingly strive for osteoblast differentiation [50].

A. Rutkovskiy et al. summarize osteogenic differentiation in their work and describe a three-stage process in which each stage is characterized by certain molecular markers. They describe how RUNX2 is essential for the proliferation and osteogenic differentiation of hMSCs. Stage two is characterized by exit from the cell cycle and serves to build up an extracellular bone matrix. In stage three, matrix mineralization occurs [51]. All these stages are mediated by expression of different proteins, 10 of which were examined in this study. These marker proteins play a crucial role in bone healing and osseointegration processes that occur after the insertion of an implant into the bone.

In order to better classify our results, we designed a schematic representation of proliferation and differentiation (Figure 5). In the following, we discuss the different marker genes examined in this paper. To enable a better understanding, we first discuss the early and then the late markers of osteoblast differentiation. The late markers were detected by us, but the temporal relationship in the early phase of bone healing must be considered. It should be noted that bone healing is a dynamic process and cellular functions take place in parallel.

RUNX 2: Runt-related transcription factor 2 (RUNX2) is an important protein that is essential for osteoblast differentiation and irreplaceable in bone structure formation [52]. Isolated evaluation of the results for RUNX2 is more difficult due to the different effects during proliferation and differentiation. On the one hand, it serves as a marker for the osteogenic cell line and regulates cell differentiation [53,54]. On the other hand, it is known that RUNX2 inhibits the cell proliferation of osteoblast progenitor cells. Its expression is inversely proportional to the speed of osteoblast proliferation; therefore, RUNX2 is thought to play an important role in controlling cell proliferation [55]. While it controls the proliferation of stem cells, it also occurs in differentiated osteoblasts. It is abundant in calcified bone [56]. We could demonstrate that the expression of RUNX2 with CT and DT was significantly increased at the early kinetic points compared to BONIT^®^. This can be explained by the fact that RUNX2 inhibits proliferation to promote cell differentiation. In connection with the results for the other differentiation markers, we assumed that the cells in BONIT^®^ were increasingly differentiated at the early stages, while the cells in the control plates initiated differentiation. This consideration is supported by the temporal course of the expression of RUNX2. In BONIT^®^, the expression increased continuously over 48 h, a sign that further immature osteoblasts had already been formed. In the control plates, the expression of RUNX2 was significantly down-regulated from 12 to 24 h. The observation is consistent with the results of J. Pratap and M. Galindo et al., who observed that the expression of RUNX2 was down-regulated when stem cells re-entered cell proliferation [55]. From this, it can be deduced that the cells in the control coatings continued to proliferate, while the cells in BONIT^®^ differentiated. The study by M. Hentschel (2013) entitled “Investigation of progenitor cells depending on decorated, as well as non-decorated implant surfaces as a growth substrate” should be critically questioned from our point of view, and we ourselves could not grow hMSC on BONIT^®^ over a longer period of time according to his method in a first series of experiments, but Hentschel’s observations support our theory of proliferation and differentiation. He was able to show that the number of cells in DT increased significantly over seven days compared to BONIT^®^. In BONIT^®^, the number of cells remained constant [34]. RUNX2 has an effect on the formation of osteopontin, osteocalcin and COL1A1 [56,57]. This effect is important for assessing the expression patterns of our results.

COL1A1: Alpha-1 type I collagen, or collagen type 1, is a fibrillar collagen and occurs mainly in the skin, tendons, fascia, bones, connective tissue, vessels and dentin [58,59]. It is the main component of the extracellular matrix and the most common type of collagen in the human organism [60]. We could prove that, after 48 h, more COL1A1 was formed in the control plates CT and DT. Since RUNX2 had an effect on the expression of COL1A1, it can be assumed that the increased expression at hour 12 had an effect on the expression levels of COL1A1 at the other kinetic points. The time course of the expression shows the increase in the expression in CT and DT. COL1A1 is increasingly secreted by early osteoprogenitor cells [53]. Our results can be explained by the fact that more precursor cells are present in CT and DT than in BONIT^®^. Collagen plays an important role in the healing of hard tissues [61]. We interpret the results in terms of increased collagenous bone matrix formation and classify the observation in the phases of proliferation and incipient differentiation where increased collagen formation is present [51]. Collagen influences the expression of alkaline phosphatase and osteonectin. This promotes mineralization. COL1A1 can inhibit cell proliferation and promote differentiation [62]. Even if the timing and the cell line were different, it should be noted that DOT GmbH published a cell differentiation study showing increased collagen synthesis at 10 days with an osteoblast cell line (hFOB1.19) [33]. Within the first 48 h, our studies with hMSC showed an opposite result.

Osseointegration is a direct connection between bone and the actual implant without intermediate connective tissue layers [63]. The formation of connective tissue is critically evaluated in implantology. Osseointegration occurs when the implant and bone are in direct contact [64]. The formation of a collagenous bone matrix may be reduced under BONIT^®^.

AP: Alkaline phosphatase (AP) consists of a group of isoenzymes [65]. They occur ubiquitously in plants and living beings [66]. In the human organism, isoenzymes originate mainly from the bone and liver [67]. Their expression is increased in osteoblastic activity [53]. In BONIT^®^, less AP was continuously produced over 48 h compared to the control plates. There was a similar temporal expression pattern as in the control coatings. The detection of expression serves as an early marker for the detection of cell differentiation [53]. M.P. Lynch et al. describe how the expression and activity of AP is promoted by collagen [62]. This observation is compatible with our results. It could be a sign that CT and DT are forming a collagenous bone matrix. These results could be explained by a faster differentiation of the cells in BONIT^®^. Taking other marker genes into account, the observation can be explained by a shorter osteoblastic-secretory phase in BONIT^®^.

ON/SPARC: Osteonectin (ON), or secreted protein acidic and rich in cysteine (SPARC), is a frequently occurring glycoprotein, which are non-collagen matrix proteins [68,69]. We were able to prove higher expression in CT and DT after 48 h. This observation also supports the formation of a bone matrix. ON can bind to calcium ions, hydroxyapatite and collagen. It is thus involved in the development of connective tissue and promotes the differentiation of osteoblasts [70]. ON is increasingly secreted at the beginning of osteoblast differentiation [71]. If our results are interpreted in the direction that differentiation in BONIT^®^ is less advanced, this contradicts late differentiation markers. Other contexts, such as mineralization, needs to be discussed here. The expression of ON combines the phase in which collagen is formed with the phase of bone mineralization [72]. Our results indicate that mineralization was beginning to be controlled in BONIT^®^. This observation supports the consideration that the cells in BONIT^®^ are subject to faster differentiation.

IGF1: Insulin-like growth factors (IGFs) are polypeptides [73]. IGF1 is similar to insulin [74]. It plays an important role in bone growth [75,76]. IGF1 initiates the differentiation of cells and can, therefore, be interpreted as an early differentiation marker [77]. We observed no significant difference in expression between the different coatings. The time course of the expression was the same for all tested coatings. IGF1 receptors decrease in favor of insulin receptors during osteoblast differentiation [77]. A study of insulin receptors would be interesting.

BMP2: Bone morphogenetic protein 2 (BMP2) belongs to the family of TGF-β proteins and is an important cytokine [78,79]. It was first described in 1965 as an important factor in the formation of bone tissue [80,81]. The expression of BMP2 is associated with bone healing [82,83]. We observed that BONIT^®^ significantly influenced the expression of BMP2. At all times, significantly more BMP2 was expressed by the cells in the BONIT^®^ sample platelets. This can be evaluated as a sign of higher differentiation activity. In an animal study, BONIT^®^ was shown to have an effect similar to BMP2 [27]. The higher BMP2 expression under BONIT^®^ influence may explain this observation. BMP2 is a key protein in the differentiation of osteoblasts [84,85,86,87]. The time course indicated that BONIT^®^ already had an influence on differentiation in the early hours and stimulated differentiation more sustainably over 48 h. In the comparative coatings CT and DT, only up to one tenth of BMP2 was expressed after 48 h. The high difference was mainly due to the rapid decrease in BMP2 expression in the control plates. This observation supports our hypothesis that cell differentiation in CT and DT is reduced in favor of cell proliferation.

OPN: The expression analysis of osteopontin also suggested faster differentiation. Osteopontin (OPN), or sialoprotein I, is a non-collagen protein of the extracellular matrix [88,89]. It is part of the basic structure of the bone matrix and is mainly formed by differentiated osteoblasts and osteocytes [53,90]. The differentiation in BONIT^®^ is further advanced. The expression increased massively after 48 h and was up to four times higher in BONIT^®^ than in the control plates. OPN can bind hydroxyapatite and is considered a promoter of mineralization [91,92]. W. Huang et al. found that the expression was increased in late stages of differentiation [53]. This supports our hypothesis of faster differentiation.

OC (OCN): Osteocalcin (OC), or bone γ-carboxylglutamic acid-containing protein (BGP), is a peptide hormone and is produced specifically by osteoblasts or odontoblasts [53,93]. Its expression is associated with bone formation [51]. OC is a component of and the second-most abundant protein in the extracellular non-collagenous bone matrix [56]. It can be measured to determine bone turnover [94]. A higher expression, which we were able to detect in BONIT^®^ at later points in time, is associated with bone augmentation. OC has a high binding capacity for hydroxyapatite and calcium [95]. It regulates the maturation of hydroxyapatite crystals [96]. Matrix mineralization is inhibited by OC [97]. It protects against over-mineralization and excessive bone growth [98]. In combination with collagen and OPN, OC leads to more stable bone growth [99,100]. We interpret our results in such a way that under BONIT^®^ there is a more stable expansion of the bone substance and there is already control over mineralization. According to A. Rutkovskiy et al., the constellation of gene expression speaks for the third phase of osteoblast differentiation [51].

RANKL: Receptor activator of NF-κB ligand (RANKL) is a protein belonging to the family of tumor necrosis factors (TNF) [101]. Primarily, it is secreted by osteoblasts [102]. When comparing the sample plates, RANKL was significantly more strongly expressed after 48 h in BONIT^®^. This was also an indication of widely differentiated osteoblasts and supports our theory that BONIT^®^ influences differentiation. RANK (receptor for RANKL) is located on the surface of osteoclast precursor cells and mature osteoclasts [103,104]. The binding of RANKL is essential for the differentiation of osteoclasts [104,105]. In this context, the expression of RANKL can also be evaluated in connection with stimulation of the osteoclasts. The interaction of osteoblasts and osteoclasts is important for the remodeling of bone substance and follows bone mineralization. RANKL can be inhibited by a soluble form of OPG [106]. DOT GmbH published a study in which the mineralization of osteoblasts was stimulated by a BONIT^®^-extract. They were able to demonstrate by means of Kossa staining that dissolved BONIT^®^ promotes mineralization [33].

OPG: Osteoprotegerin (OPG), or osteoclastogenesis inhibitory factor (OCIF), is a cytokine receptor and belongs to the family of tumor necrosis factor (TNF) receptors [107,108]. OPG is formed by osteoblasts [109]. It is a soluble glycoprotein that can bind RANKL, and OPG is also found on the surface of osteoclasts [110,111]. The differentiation and survival of osteoclasts is inhibited by OPG, and bone resorption is suppressed [112]. Our expression analyses showed a significantly higher expression of OPG on BONIT^®^. In connection with the expression of RANKL, we interpret the results as indicating incipient control catabolic and anabolic processes in bone metabolism.

The limitations of this study include the fact that expression patterns may differ in vivo when other local proteins are present and may have different effects on the investigated cells. This is because, just two minutes after implantation of a titanium implant, more than 2800 different proteins are already detectable on the surface [113]. Moreover, only the short period of 48 h was observed, which does not cover the whole process of osteoblast differentiation that usually takes a minimum of 21 days [114]. The reader should, therefore, note that only the proliferative period and the initial matrix deposition were considered in this study.

## 5. Conclusions

We have developed a new method that allows the cultivation of human mesenchymal stromal cells (hMSCs) on the soluble coating BONIT^®^ for gene expression analysis. This study describes a defined gene expression pattern of bone metabolism that may help in understanding the influence of this CaP coating on the early phase of implant osseointegration. BONIT^®^ has a significant effect on the gene expression of hMSCs, even in the early hours. To summarize our results, it can be assumed that BONIT^®^ already stimulates the differentiation behavior of hMSCs in the proliferation and matrix phase. Despite the short study period, the early development of osteocytes can be assumed.

## Figures and Tables

**Figure 1 jfb-13-00176-f001:**
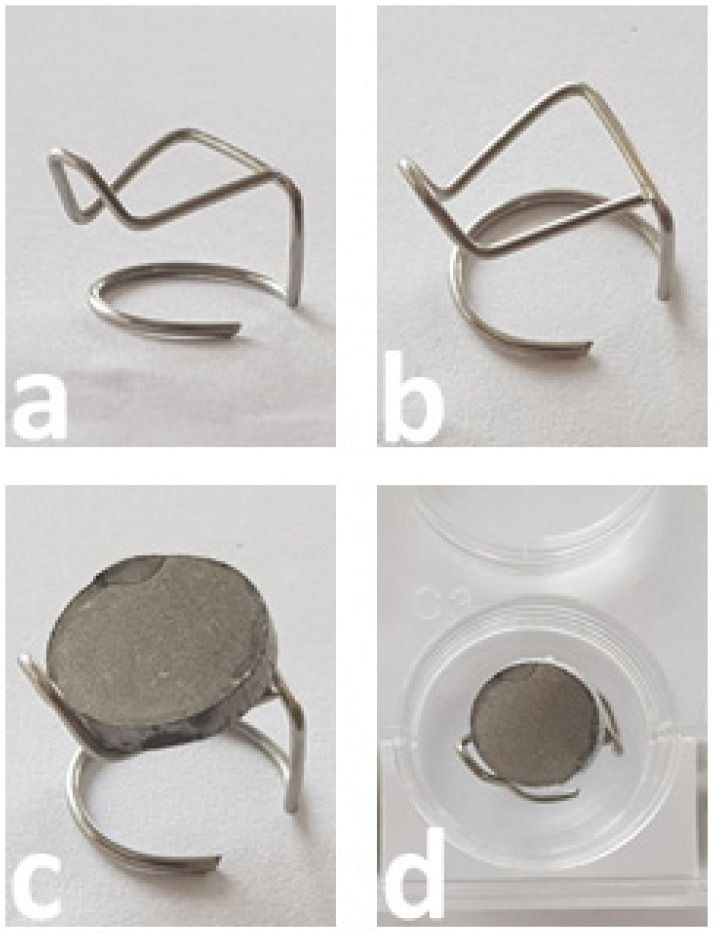
(**a**–**d**) Incubation aids bent from stainless steel wire (thickness 0.8 mm) and specially adapted to the sample plates. (**a**,**b**) The frame without plates. (**c**,**d**) The platelet (surface to be examined aligned to the ground) on the frame; the frame is located in a corrugated plate of 12 mm in diameter (**d**).

**Figure 2 jfb-13-00176-f002:**
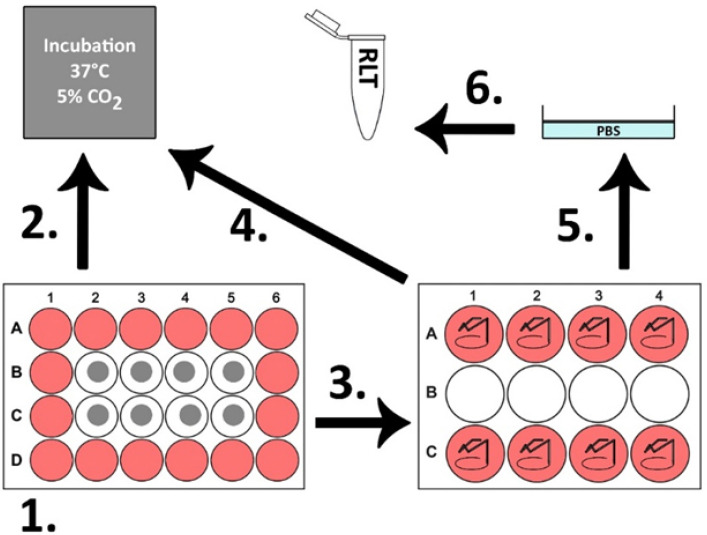
Schematic diagram of the experimental setup of the BONIT^®^ study. Step 1 = application of the cells to the sample platelets; 2 = incubation for 90 min; 3 = transfer of the sample platelets to the incubation aid; 4 = incubation for 10.5 h, 22.5 h and 46.5 h; 5 = washing of the sample platelets in PBS; 6 = Dissolution of the cells and conservation in RLT buffer, then storage at −80 °C.

**Figure 3 jfb-13-00176-f003:**
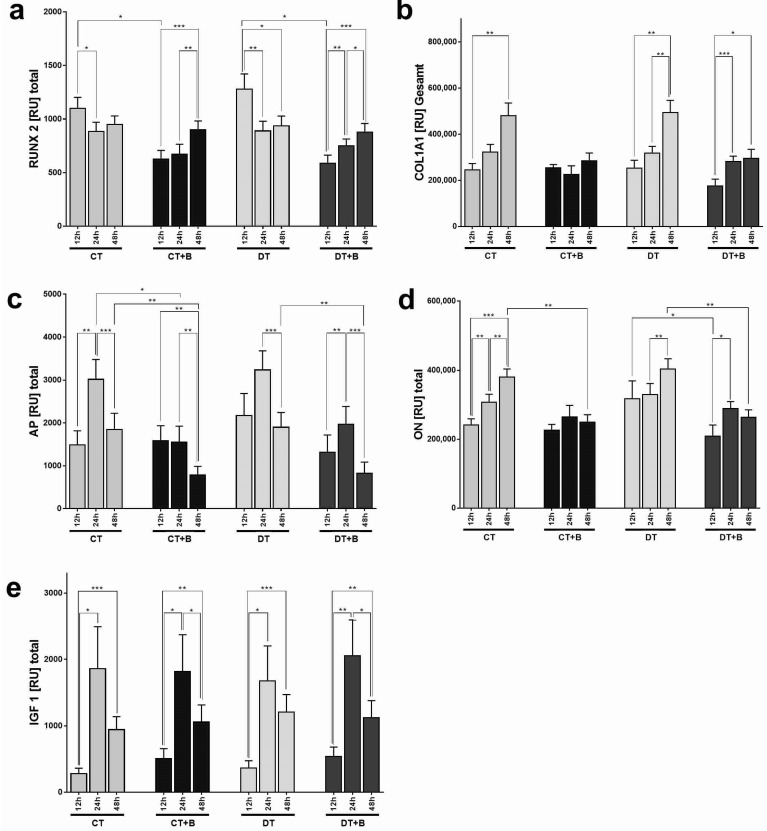
(**a**–**e**) The figures show the relative units (RU) from the gene expression analysis of runt-related transcription factor 2 (RUNX2), alpha-1 type I collagen (COL1A1), alkaline phosphatase (AP), osteonectin/secreted protein acidic and rich in cysteine (ON/SPARC) and insulin-like growth factor 1 (IGF1) for comparison of the surfaces. The figures show the mean values (+SEM, n = 10) of the expressions at three kinetic points (hours (h) = 12, 24 and 48) for the coatings CELLTex^®^ (CT), CELLTex^®^ + BONIT^®^ (CT + B), DUOTex^®^ (DT) and DUOTex^®^ + BONIT^®^ (DT + B). The levels of significance were as follows: * *p* ≤ 0.05, ** *p* ≤ 0.01, *** *p* ≤ 0.001.

**Figure 4 jfb-13-00176-f004:**
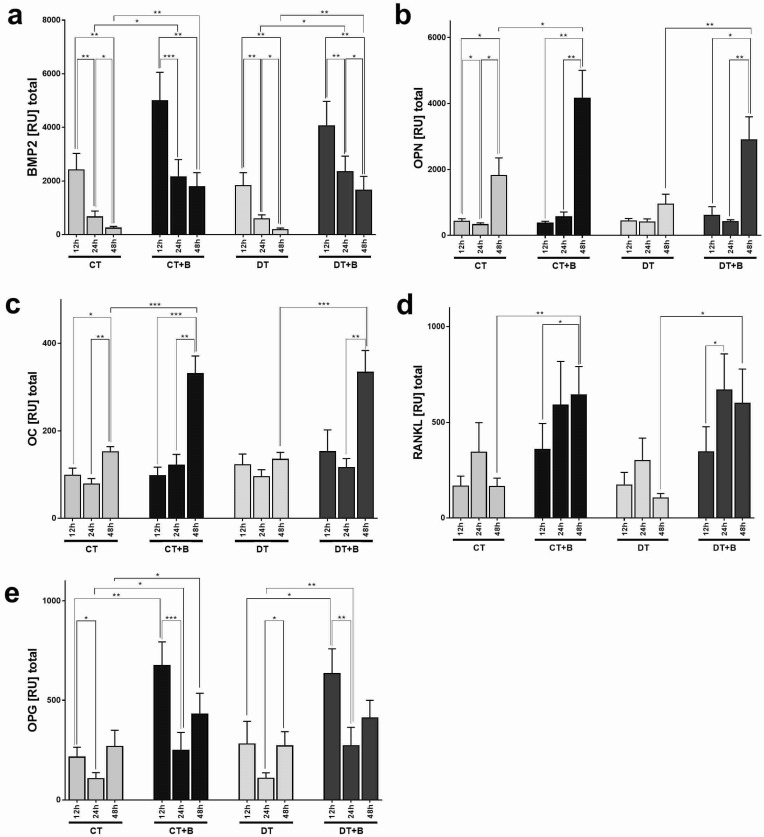
(**a**–**e**) The figures show the relative units (RU) from the gene expression analysis of bone morphogenetic protein 2 (BMP2), osteopontin (OPN), osteocalcin/bone γ-carboxylglutamic acid-containing protein (OC/BGLAP), receptor activator of NF-κB ligand (RANKL) and osteoprotegerin (OPG) for comparison of the surfaces. The figures show the mean values (+SEM, n = 10) of the expressions at three kinetic points (hours (h) = 12, 24 and 48) for the coatings CELLTex^®^ (CT), CELLTex^®^ + BONIT^®^ (CT + B), DUOTex^®^ (DT) and DUOTex^®^ + BONIT^®^ (DT + B). The levels of significance were as follows: * *p* ≤ 0.05, ** *p* ≤ 0.01, *** *p* ≤ 0.001.

**Figure 5 jfb-13-00176-f005:**
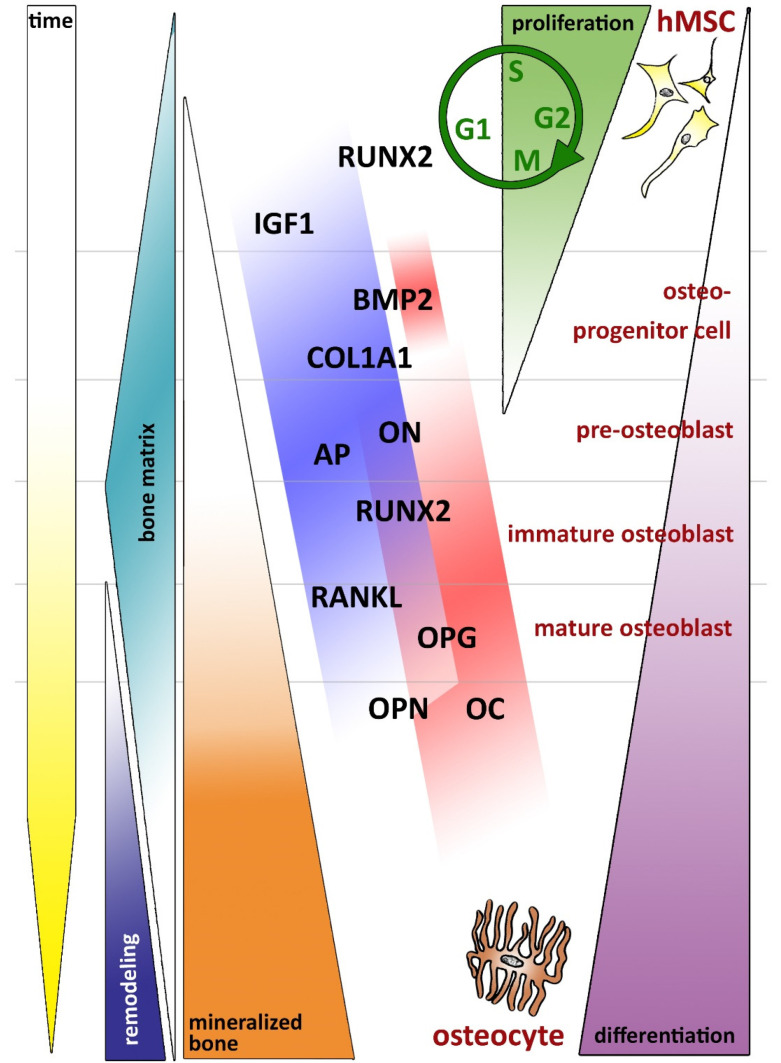
Schematic representation of proliferation and differentiation of hMSCs over time (adapted from Hentschel, M. [34]). The genes investigated in the BONIT^®^ study were integrated into a model showing the proliferation and differentiation of hMSCs and the resulting bone formation, in addition to the time course. The yellow arrow on the left side of the figure shows the time course for differentiation, on the one hand, and for bone formation, on the other hand. It should be kept in mind that both progressions have different durations and cannot be equated. The dependence of proliferation (green triangle) and differentiation (purple triangle) is shown on the right side of the figure. Proliferation is characterized by mitotic activity. Differentiation of hMSCs to osteocytes occurs through several intermediate stages. Depending on how far differentiation has progressed, mineralized bone is formed via a bone matrix (turquoise pyramid) that is subject to constant remodeling (orange and blue triangles). We show an attempt to time the cells at hour 48 in the BONIT^®^ coatings (= red) and in CT and DT (= royal blue).

## Data Availability

The data presented in this study are available on request from the corresponding author. The data are not publicly available due to privacy.

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
