# Peer review of "Influence of a Calcium Phosphate Coating (BONIT®) on the Proliferation and Differentiation Potential of Human Mesenchymal Stroma Cells in the Early Phase of Bone Healing"

_jfb, 2022, doi:10.3390/jfb13040176_

Round 1

Reviewer 2 Report

In the present paper the authors explored the influence of calcium phosphate coating on the proliferation and differentiation potential of human mesenchymal stroma cells in the early phase of bone healing.

The paper is well written, covers an important topic and I believe it would be of interest to the broad audience of the Journal of Functional Biomaterials.

However, there are some minor issues which should be addressed prior publication:

1. I would suggest to change article type from review to original paper/research article

2. minor mistakes: line 292 - wrong spelling of the word granulat

Figure 7. - spelling mistake in the word osteocyte
